# Rapid Hormetic Responses of Photosystem II Photochemistry of Clary Sage to Cadmium Exposure

**DOI:** 10.3390/ijms22010041

**Published:** 2020-12-22

**Authors:** Ioannis-Dimosthenis S. Adamakis, Ilektra Sperdouli, Anetta Hanć, Anelia Dobrikova, Emilia Apostolova, Michael Moustakas

**Affiliations:** 1Department of Botany, Faculty of Biology, National and Kapodistrian University of Athens, 15784 Athens, Greece; 2Institute of Plant Breeding and Genetic Resources, Hellenic Agricultural Organization—Demeter, Thermi, 57001 Thessaloniki, Greece; ilektras@bio.auth.gr; 3Department of Trace Analysis, Faculty of Chemistry, Adam Mickiewicz University, 61-614 Poznań, Poland; anettak@amu.edu.pl; 4Institute of Biophysics and Biomedical Engineering, Bulgarian Academy of Sciences, 1113 Sofia, Bulgaria; aneli@bio21.bas.bg (A.D.); emya@bio21.bas.bg (E.A.); 5Department of Botany, Aristotle University of Thessaloniki, 54124 Thessaloniki, Greece

**Keywords:** *Salvia sclarea*, chlorophyll fluorescence imaging, non-photochemical quenching, tolerance mechanism, photoprotective mechanism, oxidative stress, reactive oxygen species, toxicity, adaptive response, photochemical efficiency

## Abstract

Five-day exposure of clary sage (*Salvia sclarea* L.) to 100 μM cadmium (Cd) in hydroponics was sufficient to increase Cd concentrations significantly in roots and aboveground parts and affect negatively whole plant levels of calcium (Ca) and magnesium (Mg), since Cd competes for Ca channels, while reduced Mg concentrations are associated with increased Cd tolerance. Total zinc (Zn), copper (Cu), and iron (Fe) uptake increased but their translocation to the aboveground parts decreased. Despite the substantial levels of Cd in leaves, without any observed defects on chloroplast ultrastructure, an enhanced photosystem II (PSII) efficiency was observed, with a higher fraction of absorbed light energy to be directed to photochemistry (Φ*_PSΙΙ_*). The concomitant increase in the photoprotective mechanism of non-photochemical quenching of photosynthesis (NPQ) resulted in an important decrease in the dissipated non-regulated energy (Φ*_NO_*), modifying the homeostasis of reactive oxygen species (ROS), through a decreased singlet oxygen (^1^O_2_) formation. A basal ROS level was detected in control plant leaves for optimal growth, while a low increased level of ROS under 5 days Cd exposure seemed to be beneficial for triggering defense responses, and a high level of ROS out of the boundaries (8 days Cd exposure), was harmful to plants. Thus, when clary sage was exposed to Cd for a short period, tolerance mechanisms were triggered. However, exposure to a combination of Cd and high light or to Cd alone (8 days) resulted in an inhibition of PSII functionality, indicating Cd toxicity. Thus, the rapid activation of PSII functionality at short time exposure and the inhibition at longer duration suggests a hormetic response and describes these effects in terms of “adaptive response” and “toxicity”, respectively.

## 1. Introduction

Cadmium (Cd) is occurring in soils at low concentrations but can arise to high concentrations as a result of numerous human activities, while being not biodegradable in the soil, it is considered as one of the most toxic elements, and also a non-essential element for plants [1,2,3,4,5,6]. Cadmium is taken up by roots, and its translocation from the roots to the shoots and leaves with subsequent accumulation in the chloroplasts, will eventually disturb photosynthesis [7]. However, many plant species have established numerous special effective mechanisms for Cd detoxification and tolerance [4,5,6].

Cadmium tolerance is related with declines of the internal Cd accumulation [8] but some plant species are able to sustain growth or even improve their growth and functioning under Cd exposure, despite a high Cd accumulation in roots and shoots [9,10,11]. Foliar Cd content above 0.01% dry biomass (100 μg g^−1^) is considered extraordinary and a limit value for Cd hyperaccumulation [7,12].

Plants can cope with stress by a plethora of structural and functional mechanisms while low-level stress or short duration exposure stimulates plant performance [13,14,15,16,17,18,19]. This can be achieved through the involvement of a basal level of reactive oxygen species (ROS) [20,21,22,23,24,25], which are regulated by the non-photochemical quenching (NPQ) photoprotective mechanism of photosynthesis [19,22,26,27]. An elevated NPQ reduces the electron transport rate (ETR), avoiding ROS formation [28,29]. ROS generation can be a direct consequence to photosystem II (PSII) damage or can inhibit the repair of PSII reaction centers [29,30,31,32].

Dose–response studies are proposing hormesis as a central dose–response phenomenon for a variety of stressors [8,33,34,35,36,37,38]. Hormesis is a widespread phenomenon usually in nature, independent of the kind of stressor, the physiological process, or the organism it occurs [8,33,34,35,36]. It is described as the stimulatory effect of low doses or short exposure times, described by a biphasic dose–response with a low dose stimulation and a high dose inhibition [39,40,41] representing an “over-compensation” response to a disruption in homeostasis [42].

*Salvia sclarea* (clary sage) is a flowering herb that is native in the Mediterranean basin area, possessing pharmacological activities, and being traditional used as a treatment for eye health and hair tonic. It is used for pharmaceutical purposes, having antibacterial properties, while in aromatherapy, clary sage essential oil is used to alleviate stress acting as an anti-stressor, and when inhaled, it elicits feelings of relaxation and helps to reduce blood pressure [43]. *S. sclarea* is tolerant to heavy metals and has been characterized as an Zn and Cd accumulator, while its essential oils are not contaminated with heavy metals [44].

Since any substantial effect on plant growth after exposure to Cd can be detected only if photosynthesis is disturbed [45], a hormetic response to Cd is frequently coupled with changes in the mechanisms of photosynthesis or the photosynthetic apparatus [8]. The toxic effects of Cd provoke oxidative stress in plants and are related with the production of ROS [4,36,46,47] that can be regulated by NPQ in such a way so that plants can cope with the stress [19,22,26,27]. Therefore, it was hypothesized that after exposure of clary sage to Cd, the photosynthetic machinery could display a hormetic response to Cd, if plant tolerance mechanisms were activated and photosystem II photochemistry could be regulated by the photoprotective mechanism of NPQ in a such way that PSII functionality is enhanced without any chloroplast ultrastructure perturbations.

## 2. Results

### 2.1. Cadmium Accumulation and Elemental Concentrations

Upon exposure of plants to 100 μM Cd in hydroponics, Cd concentrations of aboveground tissues (shoot-leaves) and roots increased (*p* < 0.05) by 31-fold (Figure 1a) and 2900-fold (Figure 1b) respectively, with Cd ions to be retained almost exclusively in the roots and reaching 23,941 ± 715 µg g^−1^ vs. 53.3 ± 1.6 µg g^−1^ in the aboveground tissues. Cadmium exposure enhanced total Zn, Cu, and Fe uptake by 2.1-fold (Figure 2a), 1.7-fold (Figure 2b), and 1.5-fold (Figure 2c) respectively, but decreased their translocation to the leaves by 45%, 16%, and 60%, respectively. Zinc, Cu, and Fe content in roots after Cd exposure reached 542 ± 16, 112.2 ± 3.5, and 1696 ± 51 µg g^−1^ from 195.4 ± 5.8, 60.6 ± 1.8, and 945 ± 28 µg g^−1^, respectively, while in aboveground tissues from 72.9 ± 2.2, 12.24 ± 0.36, and 224.8 ± 6.7 µg g^−1^, decreased to 32.58 ± 1.6, 10.29 ± 0.28, and 88.4 ± 2.6 µg g^−1^, respectively. Total Ca uptake was significantly reduced (58%), with root Ca accumulation to remain almost unaffected, but Ca translocation to shoot-leaves to be significantly reduced (84%) (Figure 2e). Thus, after Cd exposure, Ca content in roots was 10,122 ± 304 µg g^−1^ from 10,005 ± 300 µg g^−1^ before Cd treatment, but in the aboveground tissues, decreased to 3820 ± 115 µg g^−1^ from that of 23,484 ± 704 µg g^−1^ before Cd treatment (Figure 2e). Total Mn (Figure 2d) and Mg (Figure 2f) uptake decreased (*p* < 0.05) by 5% and 49%, respectively, after Cd exposure, but while Mg accumulation decreased in both roots (43%) and shoots-leaves (53%) (Figure 2f), Mn accumulation increased in roots (1.6-fold), but its translocation to the shoots decreased (65%) after Cd exposure (Figure 2d). Manganese increased in roots from 62.01 ± 1.8 µg g^−1^ before Cd treatment to 98.1 ± 2.9 µg g^−1^, while in aboveground tissues decreased from 64.2 ± 1.9 µg g^−1^ before Cd treatment to 22.88 ± 0.69 µg g^−1^ after Cd exposure (Figure 2d). Magnesium, the next most negatively affected element after Ca, decreased in roots from 4184 ± 126 µg g^−1^ before Cd treatment to 2391 ± 72 µg g^−1^ after Cd exposure, while in aboveground tissues from 5851 ± 175 µg g^−1^ before Cd treatment to 2778 ± 83 µg g^−1^ after Cd exposure (Figure 2f).

### 2.2. Chlorophyll a and Chlorophyll b Content after Cadmium Exposure

Chlorophyll *a* (Chl*a*) content decreased (*p* < 0.05) in *S. sclarea* leaves exposed to Cd for 2 and 5 days compared to their respective controls (Figure 3a). The same response pattern was observed in chlorophyll *b* (Chl*b*) content of *S. sclarea* leaves after 2- and 5-days exposure to Cd stress (Figure 3b). These significant chlorophyll content decreases can be explained by the significantly decreased Mg uptake (Figure 2f).

### 2.3. The Efficiency of Photosystem II after Cadmium Exposure

In order to understand how PSII functionality is affected by exposure of plants to Cd, we measured the maximum efficiency of PSII photochemistry (F*v*/F*m*) (Figure 4a) and the efficiency of the water-splitting complex on the donor side of PSII (F*v*/F*o*) [29] (Figure 4b). Both parameters increased (*p* < 0.05) in *S. sclarea* plants exposed to Cd stress for 2 and 5 days compared to their respective controls, indicating an enhanced PSII functionality under Cd stress. However, this enhanced PSII functionality up to 5 days exposure had not any significant influence on plant biomass, but 8 days Cd exposure reduced whole plant biomass by 18% (*p* < 0.05).

### 2.4. Changes in the Quantum Yields and the Fraction of Open Photosystem II Reaction Centers after Cadmium Exposure under Low Light

The quantum efficiency of PSII photochemistry (Φ*_PSΙΙ_*) measured at low light (LL, 220 μmol photons m^−2^ s^–1^) (Figure 5a) increased (*p* < 0.05) in *S. sclarea* plants exposed to Cd stress for 2 and 5 days compared to their respective controls, indicating a higher fraction of absorbed light energy to be directed to photochemistry under Cd stress. The concomitant increase in the quantum yield of regulated heat dissipation in PSII (Φ*_NPQ_*) (Figure 5b) under 2- and 5-days Cd stress resulted in a significant decrease in the quantum yield of non-regulated energy dissipated in PSII (Φ*_NO_*) (Figure 5c), indicating a better use of the absorbed light energy in *S. sclarea* Cd stressed plants. The fraction of open PSII reaction centers (*q*_p_), at LL (Figure 5d), increased (*p* < 0.05) in *S. sclarea* plants exposed to Cd stress for 2 days, while at 5 days exposure remain the same, compared to controls.

### 2.5. Changes in Non-Photochemical Fluorescence Quenching and Electron Transport Rate after Cadmium Exposure under Low Light

Non-photochemical quenching (NPQ) increased (*p* < 0.05) in plants exposed to Cd stress for 2 and 5 days compared to their respective controls (Figure 5e). Electron transport rate measured at 220 μmol photons m^−2^ s^–1^ (Figure 5f) increased (*p* < 0.05) in *S. sclarea* plants exposed to Cd stress for 2 and 5 days compared to their respective controls, following the pattern of Φ*_PSΙΙ_* (Figure 5a).

### 2.6. Changes in Excess Excitation Energy under Low and High Light after Cadmium Exposure

The excess excitation energy (EXC) at PSII in *S. sclarea*, at 220 μmol photons m^−2^ s^–1^ (LL) after 2 days Cd exposure, decreased (*p* < 0.05) compared to control, while, after 5 days Cd exposure, it was at the same level with control (Figure 6a). However, under high light (HL, 900 μmol photons m^−2^ s^–1^) after 5 days Cd exposure, EXC increased (*p* < 0.05) compared to control (Figure 6b), indicating that the synergistic effect of Cd stress and HL resulted in a lower efficiency of light energy use by PSII.

### 2.7. Changes in the Quantum Yields under High Light after Cadmium Exposure

The allocation of absorbed light energy to PSII photochemistry (Φ*_PSΙΙ_*) measured at HL in *S. sclarea* plants exposed to Cd for 5 days decreased (*p* < 0.05) compared to controls, while the regulated heat dissipation (Φ*_NPQ_*) increased (*p* < 0.05) in such a degree that the non-regulated energy dissipation (Φ*_NO_*) in *S. sclarea* plants exposed to Cd for 5 days did not differ compared to controls (Figure 7).

### 2.8. Changes in Non-Photochemical Fluorescence Quenching, Electron Transport Rate, and the Fraction of Open Photosystem II Reaction Centers under High Light after Cadmium Exposure

Non-photochemical quenching (NPQ), measured at HL, increased (*p* < 0.05) *in S. sclarea* plants exposed for 5 days to Cd compared to control plants (Figure 8a), while PSII electron transport rate decreased (*p* < 0.05) compared to controls (Figure 8b), following the pattern of Φ*_PSΙI_* (Figure 7). The fraction of open PSII reaction centers (*q*_p_) decreased (*p* < 0.05) in *S. sclarea* plants exposed for 5 days to Cd compared to control plants (Figure 8c).

### 2.9. Chlorophyll a Fluorescence Images under Low and High Light

Chlorophyll *a* fluorescence images of the fluorescence parameters Φ*_PSΙΙ_* and Φ*_NO_*, measured at LL and HL, of control and 5 days Cd-treated *S. sclarea* plants, revealed a spatial heterogeneity over the whole leaf area (Figure 9). The heterogeneity was higher under Cd exposure with Φ*_PSΙΙ_* values at the center of the leaf and near the main leaf vein to have lower values compared to marginal, while the spatial heterogeneity was even higher under HL and Cd exposure (Figure 9).

The effective quantum yield of PSII photochemistry (Φ*_PSΙΙ_*) after 5 days Cd exposure, was higher under LL compared to control, but it was lower under HL compared to control (Figure 9). The non-regulated energy loss in PSII (Φ*_NO_*), under both LL and HL, was lower in 5 days Cd-treated *S. sclarea* plants compared to controls (Figure 9). At longer duration exposure (8 days) to Cd under LL, the inhibition of PSII functionality that was observed (Figure 10) resulted in the reduction of whole plant biomass by 18% (*p* < 0.05). At 8 days exposure of *S. sclarea* plants to Cd, the lowest F*v*/F*m* values were found near the midvein, while the lowest Φ*_PSΙΙ_* values were at the half leaf area near the base (Figure 10).

### 2.10. Lipid Peroxidation and Hydrogen Peroxide (H_2_O_2_) after Cadmium Exposure

The final product of lipid peroxidation, malondialdehyde (MDA) content (Figure 11b), increased with increased exposure time to Cd. The same pattern was observed in H_2_O_2_ generation (Figure 11a).

This trend was also obvious in the histochemically detected H_2_O_2_ production of *Salvia sclarea* leaves (Figure 12). After 5 days exposure to Cd, the increased H_2_O_2_ production was detected mainly in the leaf midveins near the basal leaf area, while after 8 days exposure, the highly increased H_2_O_2_ was not identified in the midveins but it was noticed to spread to the whole leaf (Figure 12).

### 2.11. Chloroplast Ultrastructure after Cadmium Exposure

Leaves from control plants exhibited rather electronically dense mesophyll chloroplasts (Figure 13a) that showed a typical internal membrane structure with well-organized grana and stroma thylakoids (Figure 13a). Chloroplasts, in both 2 days (Figure 13b) and 5 days (Figure 13c) Cd-treated plants, did not show any noticeable structural disruption having a similar appearance to the control, with the 5 day Cd-treated plastids to appear more electronically dense (Figure 13c). However, after 8 days exposure to Cd, chloroplasts appeared even more electronically dense and their thylakoids were swollen (Appendix A). In control, 2 days and 5 days Cd-treated plants, starch grains were noticeable in chloroplasts, which were absent in 8 days Cd-treated chloroplasts (Appendix A).

## 3. Discussion

Exposure of *S. sclarea* plants to 100 μM Cd for 5 days in hydroponics resulted in a high Cd uptake with a 2400-fold increase at the whole plant level but with Cd ions to be retained almost exclusively in the roots (Figure 1b) and only 53.3 µg g^−1^ to be translocated to the aboveground tissues (Figure 1a). Low Cd accumulation in leaves may represent a tolerance mechanism that protects the photosynthetic equipment against additional oxidative stress [48,49,50]. In the tolerant plant species, the excess heavy metals in roots play a significant role by sequestrating and detoxifying the extreme amount of heavy metal in order to protect the delicate aboveground photosynthetic tissues [51,52]. Since leaf Cd contents greater than 5–10 μg g^−1^ have been characterized toxic to most plants [53,54,55], it seems that *S. sclarea* could have kept Cd concentration in the photosynthetic tissues in non-toxic forms. This could be done by depositing it in the vacuoles of leaf epidermal cells [56] and/or by complexation with cellular ligands [7,55,56,57]. Hyperaccumulators can accumulate Cd to levels above 100 μg g^−1^ of shoot dry weight, without showing any toxicity symptoms [7,58]. Our results agree with those of He et al. [59] that roots of Cd tolerant plants (non-hyperaccumulators) retain considerably higher Cd concentrations than the aboveground parts, and only minor Cd is translocated to the aerial parts. In comparison to the above-ground tissues, *S. sclarea* roots showed a higher bioaccumulation ability of Cd with translocation to shoots-leaves to be restricted.

Cadmium uptake is affected by Ca levels because Cd competes for Ca channels [7,60,61] and the low Ca content of the hydroponic solution may enhance Cd uptake [62], resulting in enhanced Cd and decreased Ca in many plant species [7,61], as we also observed in *S. sclarea* experiments, with Ca being the most affected element (58% total uptake decrease, Figure 2e). Likewise, in *Oryza sativa* seedlings exposed to Cd, uptake of Ca was decreased, and Ca content in both roots and aboveground parts was significantly reduced [63]. Magnesium was the next most negatively affected element after Ca in our experiments, but low Mg status has been associated with increased Cd tolerance [64,65,66,67,68], indicating that plants regulate nutrient concentrations to mitigate Cd toxicity [68].

Antagonistic effects of Cd with Fe [50,61,69] and Zn [57,70] have been frequently reported. However, in *S. sclarea* exposed to 100 μM Cd for 5 days, total Zn, Cu, and Fe uptake increased but their translocation to the aboveground parts decreased possible due to translocation barriers. It seems that Cd uptake in *S. sclarea* is not taking place through the Fe or Zn pathway, while conditions that lead to increased Cd uptake in plants may also favor increased Fe uptake [7]. Cd treatment has been frequently mentioned that it increases Fe retention in roots but obstructs its translocation to shoots, thus reducing Fe concentrations in aboveground parts [59,71,72]. In rice, Cd has been shown to be taken up predominantly via the Mn pathway [73,74] but this was not the case in *S. sclarea*, since Mn was the less negatively affected element. 

Cadmium contamination of soil has become a serious environmental alarm as it is estimated that around 30,000 t of Cd is released annually into the environment with a consequence to the food chain and a threat to human health [75]. The use of plants for heavy metals elimination from pollutant soils and water is a technique known as phytoremediation [76]. In phytoremediation, plants that absorb heavy metals from soils and translocate them to the harvestable shoots are used for phytoextraction, while those that stabilize metal contaminants through accumulation in the root zones are used for phytostabilization [75,77]. Plant species with high bioconcentration factor but relatively low translocation factor (<1) may be considered as potential phytostabilizers [78]. *Salvia sclarea* exhibited high accumulation capacity for Cd, and by limiting its translocation from roots to shoots, it may be considered a potential phytostabilizer that can be used in heavy metal contaminated environments. Other plant species that have been proposed as Cd phytostabilizers are *Iris lactea* [75] and *Sesuvium portulacastrum* [79,80].

Despite the significant levels of Cd in leaves, a higher fraction of absorbed light energy was directed to photochemistry (Φ*_PSΙΙ_*) under 2- and 5-days Cd stress, with a concomitant increase in Φ*_NPQ_* that resulted in a significant decrease in Φ*_NO_* (Figure 5). The non-regulated energy loss in PSII (Φ*_NO_*) encompasses internal conversions and intersystem crossing, which results in singlet oxygen (^1^O_2_) creation via the triplet state of chlorophyll (^3^chl*) [29,81,82,83]. To optimize photosynthesis and growth under stressful conditions, plants have evolved a variety of mechanisms against photodamage and photoinhibition [84,85]. Non-photochemical quenching is the key photoprotective process that dissipates excess light energy as heat and protects photosynthesis [81,86,87,88,89,90]. Thus, the increased non-photochemical quenching of photosynthesis (NPQ) altered ROS homeostasis through a decreased ^1^O_2_ formation. Consequently, in *S. sclarea* plants exposed to 100 μM Cd, ROS homeostasis could be regulated by NPQ in such a way so that plants can cope with Cd stress [19,22,26,27].

The potential PSII efficiency of *S. sclarea* plants exposed to 100 μM Cd estimated by the maximum efficiency of PSII photochemistry (F*v*/F*m*) (Figure 4a) and the efficiency of the water-splitting complex on the donor side of PSII (F*v*/F*o*) (Figure 4b) [29,91] indicated an enhanced PSII functionality under Cd stress. In accordance, *S. sclarea* plants exposed to 100 μM Cd show an increased capacity to keep quinone (QA) oxidized, thus, to have a higher fraction of open PSII reaction centers (*q*_p_) compared to controls (Figure 5d). In other words, *S. sclarea* plants exposed to 100 μM Cd show a low PSII excitation pressure associated with toxicity tolerance mechanisms [92,93]. High excitation pressure defines excess energy and consequently a disproportion between energy resource and requirement [94]. This discrepancy leads to an increase in the energy transmitted from chlorophyll to oxygen, resulting in ^1^O_2_ generation [95]. Control *S. sclarea* plants that show increased excess excitation energy (EXC) at PSII (Figure 6a), show also increased ^1^O_2_ creation via ^3^chl*, compared to plants exposed to Cd for 2 days (Figure 5c). In contrast to ^1^O_2_ generation that decreased under 2 and 5 days Cd exposure (Figure 5c), H_2_O_2_ production after 5 days Cd exposure increased compared to control (Figure 11a), being detected mainly in the leaf midveins near the basal leaf area (Figure 12), while after 8 days exposure, increased more (Figure 11a) and was noticed to spread to the whole leaf (Figure 12). Thus, since ROS are formed by energy transfer (^1^O_2_) and electron transport (H_2_O_2_) simultaneously, it appears likely that their action interferes with the signaling pathways sometimes to antagonize each other. It has been frequently shown that hydrogen peroxide disperses through leaf veins to act as a long-distance molecule, triggering the stress defence response in plants [20,25,27,83,89].

A basal level of ROS is needed for optimal growth (control) [20,25], with a low increased level of ROS to be beneficial for triggering defense responses (5 days Cd exposure), and a high level of ROS (8 days Cd exposure) to be out of the boundaries and harmful to plants [20,34]. Photosystem II responses to short time Cd exposure of *S. sclarea* can be described as a hormetic response (Figure 14), representing an “over-compensation” response to a disruption in homeostasis [42].

Although excess Cd accumulation is detrimental to plants, different strategies of Cd tolerance and accumulation are adopted by plants [97]. Tang et al. [98] described a stimulation of plant growth, increase of photosynthesis, and an up-regulation of the related genes in *Sedum alfredii* exposed to 5 μM Cd. Similar results with stimulation of growth when *Noccea caerulescens* was exposed to 100 μM Cd were reported by Lombi et al. [99]. A stimulatory effect of Cd on the photosynthetic apparatus of *Arabidopsis halleri* was also described recently [33]. Małkowski et al. [36] reported a stimulation of the photosynthetic rate by Cd only at low concentrations, whereas at higher Cd concentrations, there was a significant decrease compared to controls. Nevertheless, other studies have shown a dose dependent negative impact of Cd that increases with the generation of ROS and oxidative damage and the inhibition of photosynthetic rate to follow [5,100,101,102,103,104,105,106]. On the other hand, protection to stress through ROS production [25,107] has been shown that it can be regulated by NPQ in such a way so that plants can cope with stress [22,26,27].

A negative impact of Cd on photosynthesis has been assigned to decreases in chlorophylls; ascribed to Cd-induced damage in chloroplasts’ ultrastructure [108,109]. However, in our experiment, the significant chlorophyll content decreases in *S. sclarea* leaves exposed to Cd for 2 and 5 days (Figure 3a) cannot be attributed to chloroplasts’ ultrastructure destruction but rather to the significantly decreased Mg uptake (Figure 2f). However, Mg content in the leaves after 5 days Cd exposure (2778 ± 83 µg g^−1^) remained higher than the adequate range limit (2000 µg g^−1^) [110].

Cadmium has been reported to alter chloroplast ultrastructure, reduce photosynthesis [68,106,111], and inactivate enzymes involved in CO_2_ fixation [60]. Ultrastructural changes that are observed in Cd exposed leaves of sensitive plants (thylakoid dismantling, increase of lipid droplets, etc.) [112,113,114,115] are similar to those occurring at leaf ageing [116]. In Cd tolerant species, the only ultrastructural alteration observed in leaves was a reduction of starch grains in chloroplasts [50,102,117], which may be due to disorders in the photoassimilate transport or to nutrient deficiency [50,118], a phenomenon also observed after 8 days exposure to Cd (Appendix A). In *S. sclarea* Cd-treated plants, an increase in deposited electron-dense material was observed, as reported also by Mizushima et al. [50], but no other noticeable alteration in 2- and 5-days Cd-treated plastids was detected, further consolidating the chlorophyll fluorescence imaging results. Thus, a hormetic response of PSII photochemistry to short term Cd exposure was observed, indicating an “over-compensation” response to Cd disruption in homeostasis, justifying the statement of Carvalho et al. [8] that Cd can be regarded from a toxic element, a beneficial one. Hormesis research data and data on priming (preconditioning), an expression of hormesis [16,34,119,120,121], indicate that stimulatory response detection of the low-dose or short-time exposure is highly dependent on the study strategy, including dose range and the number with duration exposure and endpoint selected [13,14,16,34,52].

Exposure of *S. sclarea* plants to a combination of Cd and high light (900 μmol photons m^−2^ s^–1^) resulted in an inhibition of PSII functionality (Figure 7 and Figure 9), while the increased NPQ (Figure 8a) was inefficient to keep the same number of open reaction centers PSII (*q*_p_, Figure 8c) compared to control plants. Dissipation of excess light energy as heat (NPQ) under environmental pressure conditions is effective only if it is regulated so as to maintain the same fraction of open reaction centers as in unstressed conditions [88,89,122,123,124,125], as was observed under low light exposure of *S. sclarea* plants to Cd, with even an increased fraction of open reaction centers to occur (Figure 5d). Thus, the combination of Cd and high light points out to Cd toxicity. The same conclusion is reached [126] at longer duration exposure (8 days) to Cd at LL, with an inhibition of PSII functionality to be observed (Figure 10).

## 4. Materials and Methods

### 4.1. Plant Material and Growth Conditions

Seeds of *Salvia sclarea* L. used for the experiments were collected from the Rose Valley (Karlovo, Bulgaria). After germination on soil in a growth room for about a month, the seedlings were transferred to pots containing continuously aerated modified Hoagland nutrient solution (described in detail before) [6]. The nutrient solution was adjusted to pH 6.0 and changed every 3 days. The growth room conditions were 24 ± 1/20 ± 1 °C day/night temperature, 14/10 h day/night photoperiod with photon flux density 200 ± 20 μmol photons m^−2^ s^−1^.

### 4.2. Cadmium Treatment

Two-month-old *S. sclarea* plants in the hydroponic culture experiments were subjected to 0 or 100 μM Cd (as 3CdSO_4_ 8H_2_O) for a period up to five days. The pots containing only Hoagland nutrient solution served as the control, while all solutions were renewed every two days.

### 4.3. Determination of Elemental Concentration by Inductively Coupled Plasma Mass Spectrometry (ICP-MS)

After 5 days treatment with 0 (control) or 100 μM Cd, *Salvia* plants were harvested, separated in roots and aboveground (shoots-leaves) tissues, washed three times in deionized water, and then dried at 65 °C to constant biomass, milled and finally sieved. Dried sieved samples of 0.3 g were transferred in 10 mL quartz vessels with 65% (*v*/*v*) nitric acid (Suprapur, Merck, Darmstadt, Germany) and 30% (*v*/*v*) hydrogen peroxide (Suprapur, Merck, Darmstadt, Germany) in 3:1 ratio. Digestion was carried out in the microwave assisted digestion system Ethos One (Milestone Srl, Sorisole, BG, Italy). The process run out in 3 stages: ramp time—20 min to reach 200 °C and 1500 W; hold time—30 min at 200 °C and 1500 W; cooling—30 min. The next step was the quantitative transfer of digested samples into polypropylene tubes and dilution with demineralized water (Direct-Q 3 UV, Merck, Darmstadt, Germany). All prepared samples were diluted immediately prior to inductively coupled plasma mass spectrometer (ICP-MS) analysis. Samples were analyzed in an ICP-MS model ELAN DRC II (PerkinElmer Sciex, Toronto, Canada) [127]. ICP-MS operational conditions, instrumental settings calibration solutions, data validation, and validation parameters are given in Appendix B. Elemental analysis was performed for Cd, Cu, Ca, Mg, Mn, Fe, and Zn.

### 4.4. Measurements of Chlorophyll a and Chlorophyll b Content

Chlorophyll *a* (Chl*a*) and chlorophyll *b* (Chl*b*) content was determined according to Lichtenthaler [128]. Leaf tissue (50 mg) was homogenized with 10 mL ice-cold 80% (*v*/*v*) acetone and centrifuged at 5000× *g* for 5 min at 4 °C. The absorbance of the supernatant was measured at 646.8 and 663.2 nm (Specord 210 Plus, Ed. 2010, Analytik Jena AG, Jena, Germany) and Chl*a* and Chl*b* content was estimated from the equations: Chl*a* = 12.25 A_663.2_ − 2.79 A_646.8_; Chl*b* = 21.50 A_646.8_ − 5.10 A_663.2_ [128]. The mean values were averaged from three independent treatments with 2 repetitions for each treatment and are presented as mg g^−1^ FW.

### 4.5. Chlorophyll Fluorescence Imaging Analysis

Chlorophyll fluorescence measurements were conducted on dark adapted (20 min) leaves of *S. sclarea* plants, treated for 2 and 5 days with 0 (control) or 100 μM Cd, using an Imaging PAM M-Series system (Heinz Walz Instruments, Effeltrich, Germany) as described in detail previously [129]. Two light intensities were used for measurements of photosynthetic efficiency of *S. sclarea* leaves, a LL, similar to the growth light (220 μmol photons m^−2^ s^−1^), and a HL (900 μmol photons m^−2^ s^−1^). In each leaf, representative areas of interest (AOIs) were selected so as to have measurements of the whole leaf area. The definitions of the five main chlorophyll fluorescence parameters (Fo, Fm, Fo′, Fm′, and Fs) measured by the Imaging PAM M-Series system are presented in Appendix A, while a typical modulated fluorescence trace showing how the main five parameters are formed is presented in Appendix A. The chlorophyll fluorescence parameters calculated from the five main parameters with their definitions are described in Table 1. Representative results are also shown as color-coded images of F*v*/F*m* after dark adaptation and of Φ*_PSΙΙ_* and Φ*_NO_*, after 5 min illumination with 220 μmol photons m^–2^ s^–1^ (LL) or/and 900 μmol photons m^–2^ s^–1^ (HL).

### 4.6. Determination of Oxidative Damage

Leaf samples were frozen in liquid nitrogen and stored at −80 °C for analysis of hydrogen peroxide (H_2_O_2_) and malondialdehyde (MDA) content. The level of lipid peroxidation in *S. sclarea* leaves of control, and 5- and 8-days Cd-treated plants was measured as malondialdehyde (MDA) content determined by the reaction with 2-thiobarbituric acid (TBA), according to the method of Hodges et al. [130]. Hydrogen peroxide (H_2_O_2_) was extracted by homogenization with 50 mM K-phosphate buffer pH (6.5) and determined as described by Hossain et al. [131] after reaction with 0.1% TiCl_4_ in 20% H_2_SO_4_.

The histochemically detection of H_2_O_2_ in leaves was performed as described by Daudi and O’Brien [132] by staining with 1% 3,3′-diaminobenzidine (DAB) solution. DAB is oxidized by H_2_O_2_ in the presence of some heme-containing proteins to generate a dark brown precipitate. This precipitate is exploited as a stain to detect the presence and distribution of hydrogen peroxide in plant tissues.

### 4.7. Leaf Ultrastructure Observations by Transmission Electron Microscopy

In order to study leaf ultrastructure alterations after 2 and 5 days of Cd treatment, leaves from both Cd-treated and untreated plants were excised and segmented with a razor blade into small pieces of 0.5 × 1 mm. Leaf segments were fixed with 2% paraformaldehyde plus 4% glutaraldehyde, in 0.05 M sodium cacodylate buffer, pH 7.0 solution [129]. After a 5 h fixation at room temperature, the samples were washed with a 0.05 M sodium cacodylate buffer and post-fixed for another 3 h in a similarly buffered 2% osmium tetroxide solution (Agar Scientific, Essex, UK). Afterwards, samples were dehydrated in an acetone series, treated with propylene oxide, and embedded in Durcupan ACM resin (Fluka Chemie AG, Buchs, Switzerland). Ultrathin sections (80–90 nm) were cut in a ULTROTOME III TYPE 8801A ultramicrotome (LKB, Stockholm, Sweden), equipped with a glass knife, collected on nickel grids. The sections were stained with 2% uranyl acetate and 1% lead citrate and examined in a JEOL JEM 1011 (JEOL, Tokyo, Japan) TEM, equipped with a Gatan ES500W (Gatan, Pleasanton, CA, USA) digital camera. Digital electron micrographs were obtained with the DigitalMigrograph 3.11.2 (Gatan, Pleasanton, CA, USA) software according to the manufacturer’s instructions.

### 4.8. Statistical Analyses

Mean values were calculated from three independent treatments (biological replicates). Statistically significant differences among the means were determined using one-way analysis of variance or two-way ANOVA. Means (±SD) were considered statistically different at a level of *p* < 0.05.

## 5. Conclusions

Although surplus Cd accumulation is detrimental to most plants, different strategies of Cd tolerance and accumulation are adopted by different plant species [97]. When clary sage was exposed to Cd for a short time, tolerance mechanisms were triggered, with PSII photochemistry to be enhanced, without any defects to chloroplasts, as observed by transmission electron microscopy (Figure 13). However, exposure to a combination of Cd and high light (Figure 7 and Figure 9), or longer duration exposure to Cd alone (8 days), resulted in an inhibition of PSII functionality (Figure 10) and [126], pointing out to Cd toxicity. Thus, an activation of PSII function at short time exposures and an inhibition at longer duration suggests a hormetic response (Figure 14), and describes these effects in terms of “adaptive response” and “toxicity”, respectively.

## Figures and Tables

**Figure 1 ijms-22-00041-f001:**
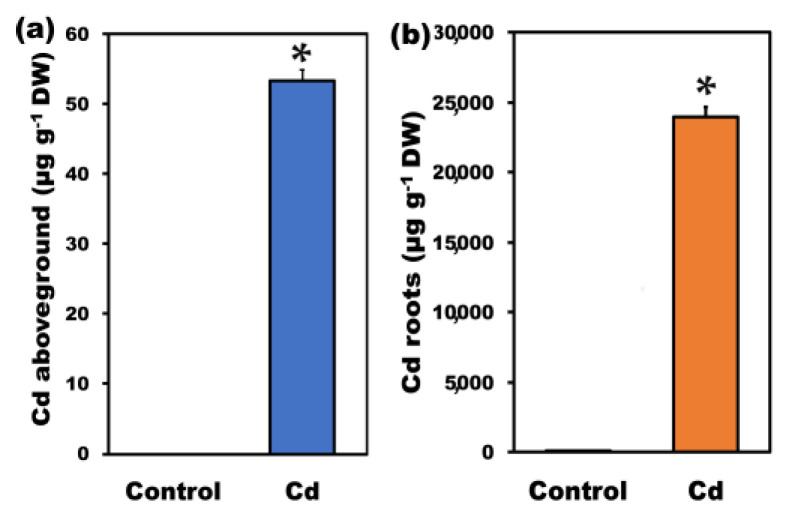
Changes of Cd accumulation in aboveground (shoots-leaves) tissues (**a**) and roots (**b**), in µg g^−1^ dry weight, after 5 days Cd treatment of *Salvia sclarea* plants. Error bars are standard deviations (*n* = 5). Means between the two treatments that are statistically different (*p* < 0.05) are marked by an asterisk (*).

**Figure 2 ijms-22-00041-f002:**
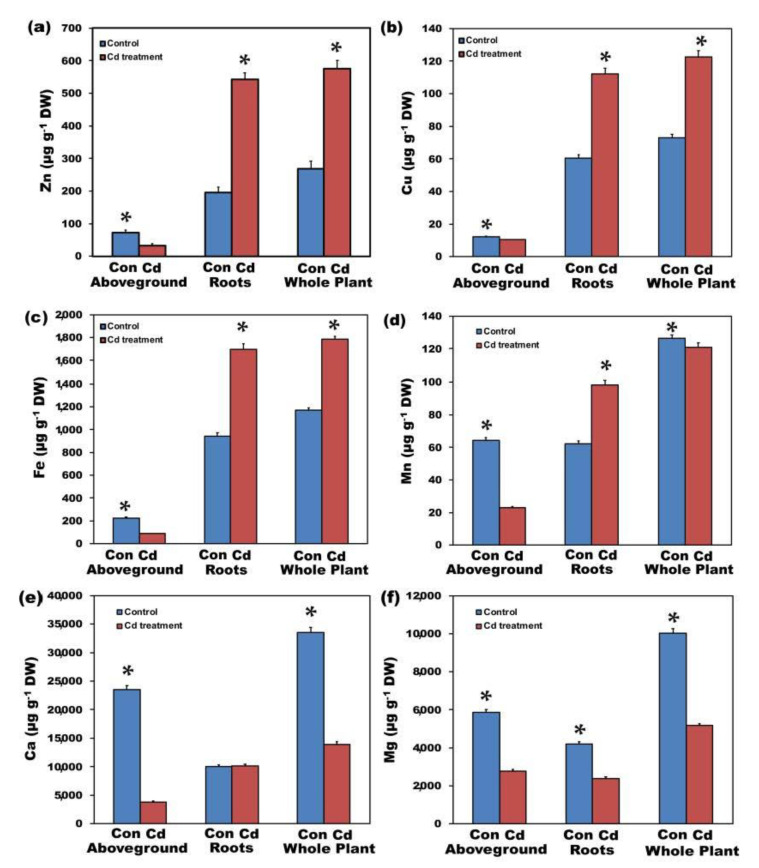
Zinc (**a**), Cu (**b**), Fe (**c**), Mn (**d**), Ca (**e**), and Mg (**f**) content, in µg g^−1^ dry weight, of control (con) and 5 days Cd-treated *Salvia sclarea* aboveground (shoots-leaves) tissues, roots, and whole plants. Error bars are standard deviations (*n* = 5). Means between the two treatments that are statistically different (*p* < 0.05) are marked by an asterisk (*).

**Figure 3 ijms-22-00041-f003:**
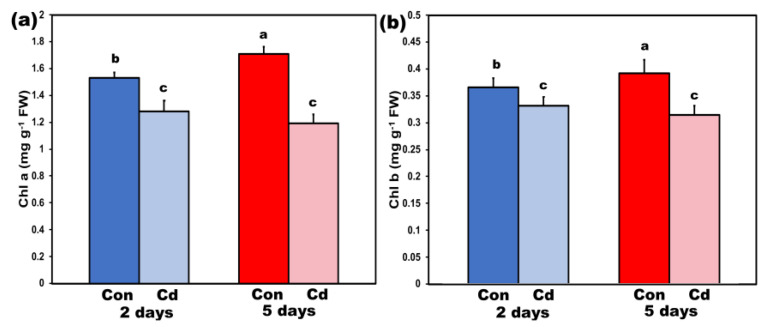
Chlorophyll *a* content (**a**) and chlorophyll *b* content (**b**), in mg g^−1^ fresh weight, of control (con) and 2- and 5-days Cd-treated *Salvia sclarea* plants. Error bars are standard deviations (*n* = 6). Columns with different letters are statistically different (*p* < 0.05).

**Figure 4 ijms-22-00041-f004:**
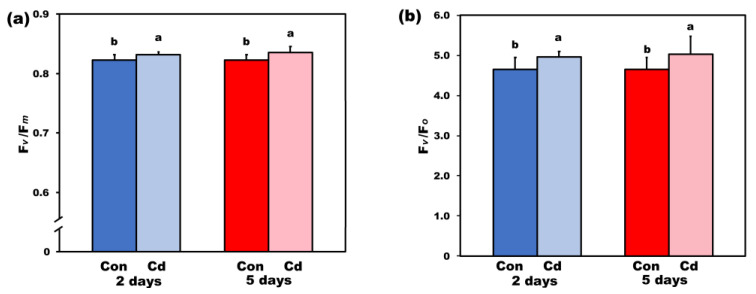
The maximum efficiency of photosystem II (PSII) photochemistry (F*v*/F*m*) (**a**), and the efficiency of the water-splitting complex on the donor side of PSII (F*v*/F*o*) (**b**), of control (con) and 2- and 5-days Cd-treated *Salvia sclarea* plants. Error bars are standard deviations (*n* = 6). Columns with different letters are statistically different (*p* < 0.05).

**Figure 5 ijms-22-00041-f005:**
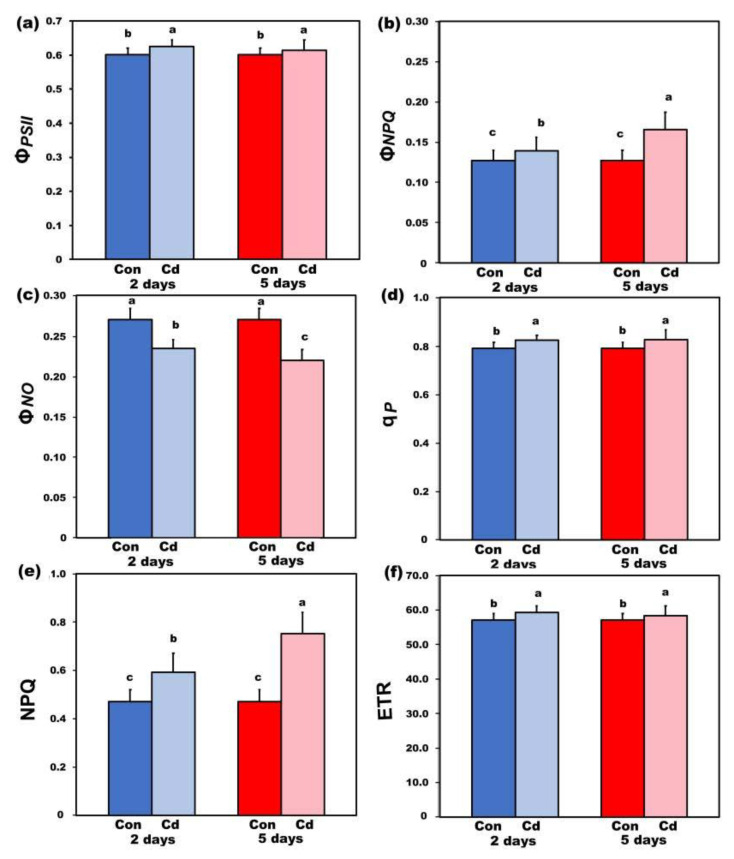
Changes in the quantum efficiency of PSII photochemistry (Φ*_PSΙΙ_*)(**a**), the quantum yield of regulated heat dissipation in PSII (Φ*_NPQ_*) (**b**), the quantum yield of non-regulated energy dissipated in PSII (Φ*_NO_*) (**c**), the fraction of open PSII reaction centers (*q*_p_) (**d**), the non-photochemical quenching (NPQ) (**e**) and the electron transport rate (ETR) (**f**), measured at 220 μmol photons m^−2^ s^–1^; of control (con) and 2- and 5-days Cd-treated *Salvia sclarea* plants. Error bars are standard deviations (*n* = 6). Columns with different letters are statistically different (*p* < 0.05).

**Figure 6 ijms-22-00041-f006:**
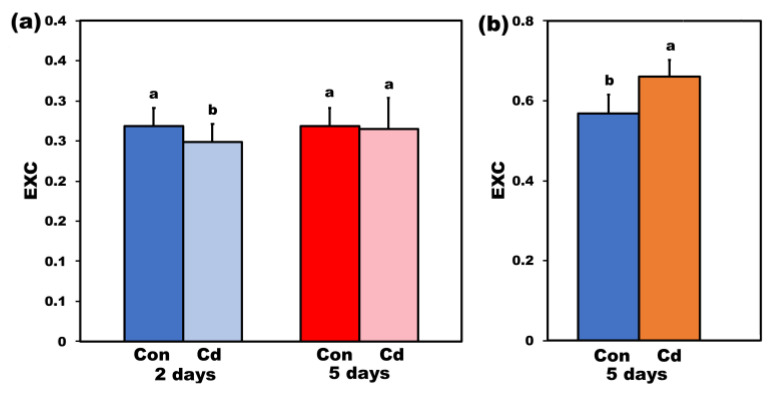
Changes in the excess excitation energy (EXC) measured at 220 μmol photons m^−2^ s^–1^ of control (con) and2- and 5-days Cd-treated *Salvia sclarea* plants (**a**) and the EXC measured at 900 μmol photons m^−2^ s^–1^ of control (con) and 5 days Cd-treated *S. sclarea* plants (**b**). Error bars are standard deviations (n = 6). Columns with different letters are statistically different (*p* < 0.05).

**Figure 7 ijms-22-00041-f007:**
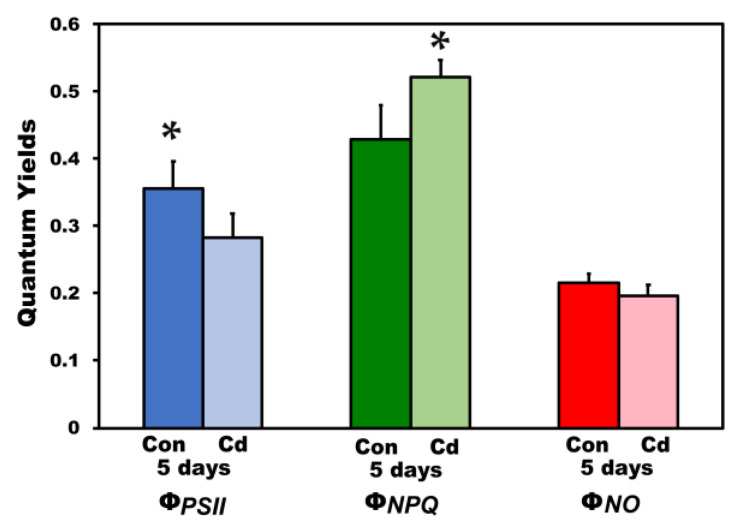
The quantum yields of PSII photochemistry (Φ*_PSΙΙ_*), the regulated heat dissipation (Φ*_NPQ_*), and the non-regulated energy dissipation (Φ*_NO_*), measured at 900 μmol photons m^−2^ s^–1^; of control (con) and 5 days Cd-treated *Salvia sclarea* plants. Error bars are standard deviations (*n* = 6). Means between the two treatments that are statistically different (*p* < 0.05) are marked by an asterisk (*).

**Figure 8 ijms-22-00041-f008:**
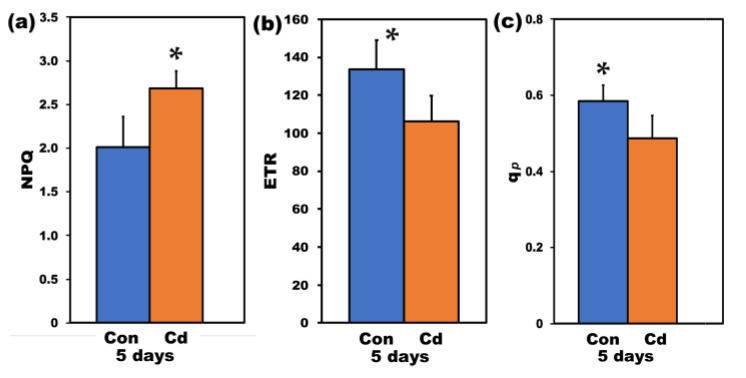
The non-photochemical fluorescence quenching (NPQ) (**a**), the relative PSII electron transport rate (ETR) (**b**), and the relative reduction state of Q*_A_*, reflecting the fraction of open PSII reaction centers (*q*_p_) (**c**), measured at 900 μmol photons m^−2^ s^–1^; of control (con) and 5 days Cd-treated *Salvia sclarea* plants. Error bars are standard deviations (*n* = 6). Means between the two treatments that are statistically different (*p* < 0.05) are marked by an asterisk (*).

**Figure 9 ijms-22-00041-f009:**
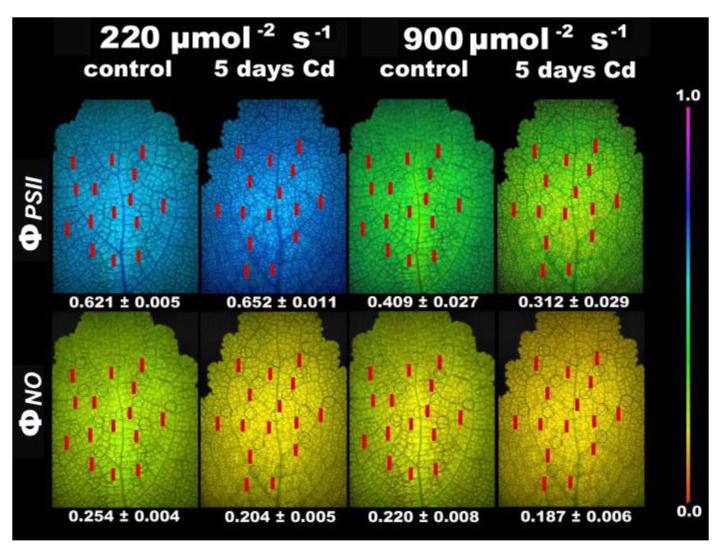
Chlorophyll fluorescence images of Φ*_PSΙΙ_* and Φ*_NO_* (measured at 220 μmol photons m^−2^ s^–1^ and 900 μmol photons m^−2^ s^–1^) of control and 5 days Cd-treated *Salvia sclarea* plants. The color code depicted at the right-side ranges from values 0.0 to 1.0. The fifteen circles in each image denote the areas of interest (AOI) that are complemented by red labels with the values of the fluorescence parameter, while whole leaf value is presented.

**Figure 10 ijms-22-00041-f010:**
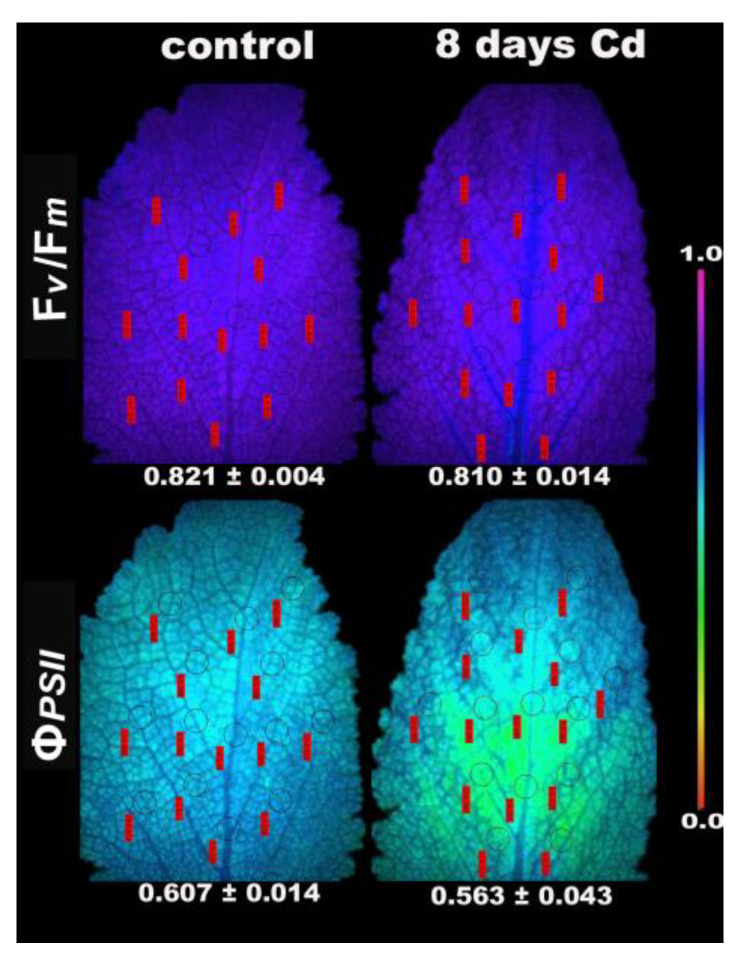
Representative chlorophyll fluorescence images of the maximum efficiency of PSII photochemistry (F*v*/F*m*), and the effective quantum yield of PSII photochemistry (Φ*_PSΙΙ_*) (measured at 220 μmol photons m^−2^ s^–1^), of *S. sclarea* leaves from control and 8 days Cd-treated plants. The color code depicted at the right-side ranges from values 0.0 to 1.0. The fourteen circles in each image are the areas of interest (AOI) complemented by red labels with the values of the fluorescence parameter. The average value of each photosynthetic parameter of the whole leaf is presented.

**Figure 11 ijms-22-00041-f011:**
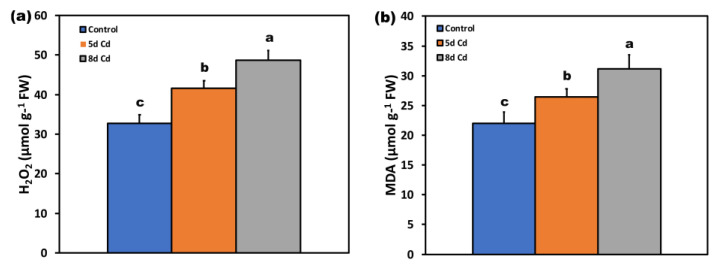
Changes in hydrogen peroxide (H_2_O_2_) generation (**a**), and lipid peroxidation production (**b**), in the leaves of *Salvia sclarea* control (con), and 5- and 8-days Cd-treated plants. Error bars are standard deviations (*n* = 6). Columns with different letters are statistically different (*p* < 0.05).

**Figure 12 ijms-22-00041-f012:**
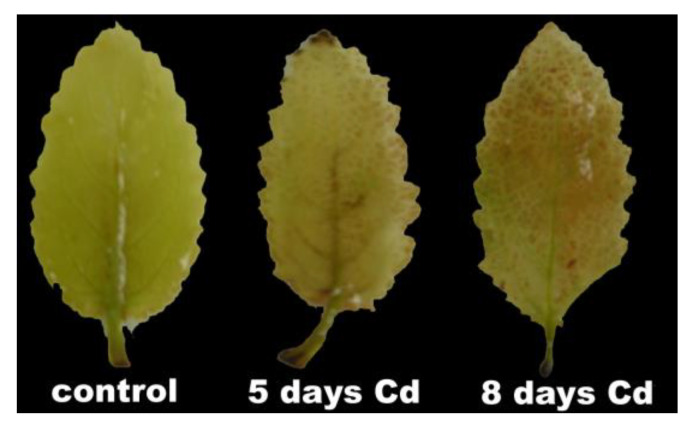
Histochemically detected H_2_O_2_ in leaves of *Salvia sclarea*, control, and 5- and 8-days Cd-treated plants. Hydrogen peroxide is forming brown precipitates with 3,3′-diaminobenzidine (DAB).

**Figure 13 ijms-22-00041-f013:**
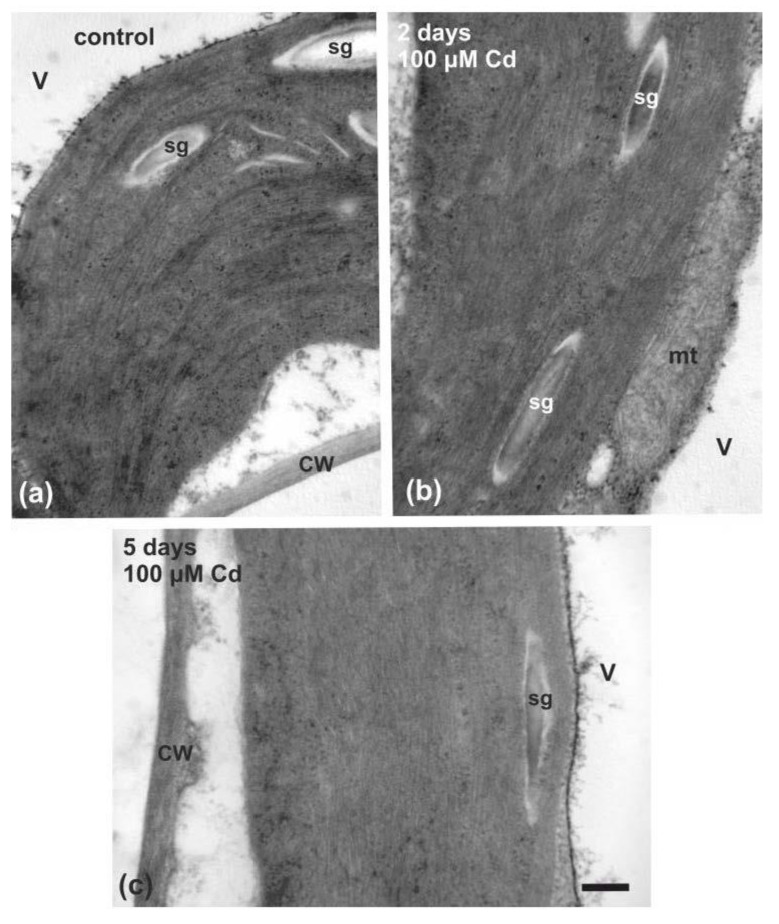
Transmission electron microscopy (TEM) images of control (untreated) chloroplasts (**a**) and 2 day (**b**), or 5 day (**c**), Cd-treated *Salvia sclarea* leaves. Chloroplasts appear electronically dense and upon Cd treatment (**b**,**c**), no noticeable disruption has been detected. cw: cell wall; mt: mitochondria; sg: starch grain; v: vacuole. Scale bar: 500 nm.

**Figure 14 ijms-22-00041-f014:**
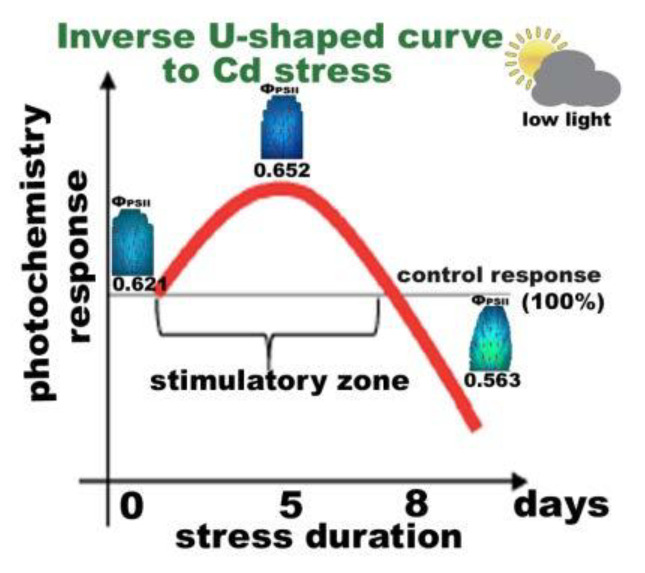
Overview of the hormetic response of photosystem II photochemistry to Cd exposure. Hormesis [96] is defined as the stimulatory effect of short exposure times of toxic constituents, e.g., Cd on a biological factor (photosystem II photochemistry), of a particular organism (*S. sclarea*). The hormetic effect is defined by an inverse U-shaped biphasic curve [8,34] in which short exposure time has a stimulatory effect; however, at longer exposure time, a toxic effect is evident.

**Table 1 ijms-22-00041-t001:** Definitions of the chlorophyll fluorescence parameters calculated from the five main chlorophyll fluorescence parameters listed in Appendix A.

Parameter	Definition	Calculation
F*v*/F*m*	Maximum efficiency of PSII photochemistry	Calculated as (Fm − Fo)/Fm
F*v*/F*o*	Efficiency of the water-splitting complex on the donor side of PSII	Calculated as (Fm − Fo)/Fo
F*v*′/F*m*′	The efficiency of open PSII reaction centers	Calculated as (Fm′ − Fo′)/Fm′
Φ*_PSII_*	The effective quantum yield of PSII photochemistry	Calculated as (Fm′ − Fs)/Fm′
*q* _p_	The photochemical quenching, that is the redox state of the plastoquinone pool, is a measure of the number of open PSII reaction centers	Calculated as (Fm′ − Fs)/(Fm′ − Fo′)
NPQ	The non-photochemical quenching that reflects heat dissipation of excitation energy	Calculated as (Fm − Fm′)/Fm′
ETR	The relative PSII electron transport rate	Calculated as ΦPSII × PAR × c × abs, where PAR is the photosynthetically active radiation c is 0.5, and abs is the total light absorption of the leaf taken as 0.84
Φ*_NPQ_*	The quantum yield of regulated non-photochemical energy loss in PSII, that is heat dissipation for photoprotection	Calculated as Fs/Fm′ − Fs/Fm
Φ*_NO_*	The quantum yield of non-regulated energy loss in PSII	Calculated as Fs/Fm
EXC	Excess excitation energy	Calculated as (Fv/Fm − ΦPSII)/(Fv/Fm)

## Data Availability

The data presented in this study are openly available in [repository name e.g., FigShare] at [doi], reference number [reference number].

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
