# Peer review of "Rapid Hormetic Responses of Photosystem II Photochemistry of Clary Sage to Cadmium Exposure"

_ijms, 2020, doi:10.3390/ijms22010041_

Round 1
Reviewer 1 Report
In this study, the authors examined the response of photosystem II (PSII) to cadmium (Cd) treatment in a Cd accumulator plant, clary sage. This study indicates that PSII functionality and ultrastructure of chloroplasts are maintained under short-term Cd treatment in the leaves of clary sage. Although this study provides useful information as a physiological research, I regret to say it only describes the response and failed to dissect the mechanisms of Cd tolerance of the plant. The data presented are clear, but the conclusion is based solely on chlorophyll fluorescence data, which is not very convincing. The major problems concerning this manuscript are as follows.
1) The authors claim that PSII phytochemistry shows hermetic response to Cd exposure. Their conclusion is based on the finding that ΦPSII is increased by Cd treatment for 5 days. It indicates enhanced PSII efficiency indeed, but it is not shown whether this increase contributes to the performance of the plant. Therefore, it is not clear whether this response has a physiological significance in clary sage.
2) Another basis of the authors’ conclusion of hermetic response is the decrease in ΦPSII by 8-day treatment which is shown in Figure S2. It is not appropriate to deduce a conclusion from non-essential (supplementary) data. I also point out that response of PSII by Cd treatment for longer duration after day 8 should be examined.
3) In Abstract (L28-29) and discussion (L274-275), the authors mention that production of reactive oxygen species (ROS) is suppressed by increased non-photochemical quenching. But this is only a speculation and is not evidenced by the data. Since ROS level can be measured, the changes in ROS level in this plant should be examined. In addition, oxidative injuries can also be examined.
4) In this study, the Cd concentration is fixed at 100 µM. I understand exposure to 100 µM Cd caused substantial accumulation of Cd in leaves, but the dose response of the plant to Cd should be examined in order to clarify whether higher concentration of Cd can cause inhibition of PSII.
I would also point out following minor issues.
5) I think references of 140 literatures are too many for a regular research paper.
6) In Materials and Methods, the information of manufacturer of electron microscope, digital camera, and DigitalMigrograph software are lacking (L392-394).
Author Response
In this study, the authors examined the response of photosystem II (PSII) to cadmium (Cd) treatment in a Cd accumulator plant, clary sage. This study indicates that PSII functionality and ultrastructure of chloroplasts are maintained under short-term Cd treatment in the leaves of clary sage. Although this study provides useful information as a physiological research, I regret to say it only describes the response and failed to dissect the mechanisms of Cd tolerance of the plant. The data presented are clear, but the conclusion is based solely on chlorophyll fluorescence data, which is not very convincing. The major problems concerning this manuscript are as follows.
We have included new data in our manuscript according to your advice so as the analysis of tolerance mechanism not to be solely based on chlorophyll fluorescence data. An analysis of PSII tolerance mechanism to Cd exposure is given on lines 300-320.
1) The authors claim that PSII phytochemistry shows hermetic response to Cd exposure. Their conclusion is based on the finding that ΦPSII is increased by Cd treatment for 5 days. It indicates enhanced PSII efficiency indeed, but it is not shown whether this increase contributes to the performance of the plant. Therefore, it is not clear whether this response has a physiological significance in clary sage.
We have included in the manuscript (lines 121-123, and 201-202) data referring to plant growth performance. Cd exposure up to 5 days had not any negative or positive significant influence on plant biomass, but 8 days Cd exposure reduced whole plant biomass by 18% (p<0.05).
2) Another basis of the authors’ conclusion of hermetic response is the decrease in ΦPSII by 8-day treatment which is shown in Figure S2. It is not appropriate to deduce a conclusion from non-essential (supplementary) data. I also point out that response of PSII by Cd treatment for longer duration after day 8 should be examined.
We have included the supplemental Figure S2 in the main text as Figure 9. Responses of PSII to longer duration exposure with toxicity symptoms have been include in another manuscript (Reference 127) that has been submitted for publication and it is cited in the present manuscript.
3) In Abstract (L28-29) and discussion (L274-275), the authors mention that production of reactive oxygen species (ROS) is suppressed by increased non-photochemical quenching. But this is only a speculation and is not evidenced by the data. Since ROS level can be measured, the changes in ROS level in this plant should be examined. In addition, oxidative injuries can also be examined.
We have included in the manuscript data about ROS level and oxidative stress [H2O2 content, lipid peroxidation (MDA content), and histochemical detection of H2O2].
In the abstract we don’t mention that production of ROS is suppressed by NPQ but we mention that NPQ changes ROS homeostasis and exactly “NPQ resulted in an important decrease in the dissipated non-regulated energy (ΦNO), modifying the homeostasis of reactive oxygen species (ROS), through a decreased singlet oxygen (1O2) formation.” This was not a speculation but it was documented by the presented ΦNO results. Yet, we added in the abstract what it was written in the discussion, that Cd exposure increased the basal ROS level (it is documented by the new data).
You were right about what we have written in the discussion section (L274-275) about decreased ROS. It was mistake. We have changed this sentence to fit with what it was written in the Abstract and with the new data that we included. Thank you for pointing this.
4) In this study, the Cd concentration is fixed at 100 µM. I understand exposure to 100 µM Cd caused substantial accumulation of Cd in leaves, but the dose response of the plant to Cd should be examined in order to clarify whether higher concentration of Cd can cause inhibition of PSII.
The Cd concentration used in this experiment (100 μM) was based on our preliminary experiments that show that sage plants could not tolerate higher Cd concentrations in hydroponic culture. This concentration corresponds to sublethal but toxic concentrations that can be found in heavily polluted sites, where non-tolerant plants would not grow, but hyper-tolerant species survive.
I would also point out following minor issues.
5) I think references of 140 literatures are too many for a regular research paper.
We omitted 12 references, but we added 3 new one for the 3 new methods that we added, plus another reference that was asked by reviewer No 2.
6) In Materials and Methods, the information of manufacturer of electron microscope, digital camera, and DigitalMigrograph software are lacking (L392-394).
All this information has been included in the revised manuscript
Reviewer 2 Report
The authors report hermetic effects of Cd on metal homeostasis and photosynthesis in clary sage (Salvia sclarea). I have some comments on the manuscript by Adamakis et al as follows:
(1) One of the purpose of this study is to understand the physiological responses of clary sage, which is tolerant to heavy metals, to a high concentration of Cd, so the title should include the name of plants used in this study, namely, Salvia sclarea (clary sage).
(2) Although Fig. 2 and Fig. 3 show the same kind of analysis for several metal accumulation in clary sage plants, why the authors split these data into two figures? I would like to request to combine them into one figure to make the data easy to compare.
(3) As for Figs. 4-6 and Figs. 7-9, I suggest the same thing (the data can be combined into one or two figures) as pointed out earlier for Fig. 2 and Fig. 3.
(4) One of the most important findings of this study should be the hermetic responses of PSII photochemistry to Cd exposure in clary sage as be in the title. However, the negative effects of longer (8-d) Cd exposure on PSII functionality is only shown in the Fig. S2. Moreover, only the pseudocolor images of Fv/Fm and YII are shown without any quantitative data. The authors should show data for 8-d Cd exposure like as the data for 2-d and 5-d Cd exposure.
(5) The authors show that NPQ is increased in short-term Cd treated plants, but no underlying mechanism is investigated and discussed. NPQ can be decomposed mainly into 3 processes, namely qE, qT (or qZ) and qI. Particularly, under stress conditions, distinguishment of energy-dependent qE and photoinhibitory qI is important for understanding which NPQ mechanism is mainly operated under a specific condition. Because the authors claim the importance of NPQ in the Cd-induced hermetic response of PSII photochemistry in clary sage, which actually consists fundamental parts of their study, this kind of data is necessary to understand the underlying mechanisms for the hermetic responses.
(6) No data for growth data (plant size, fresh weight or dry weight) are shown in the manuscript, but these data are very important to consider a relationship between growth and photosynthetic activity under the stress conditions.
(7) Although the authors state that no noticeable disruption was detected in chloroplast ultrastructure, but how was the content of photosynthetic pigments (chlorophyll and carotenoids)? Chlorophyll fluorescence parameters only show the efficiency and do not tell us the total photosynthetic activity. In this context, profiles of pigment content, particularly chlorophyll content because magnesium decreased by the Cd exposure, during the Cd exposure are necessary to estimate functionality of chloroplasts under Cd-exposed stresses.
(8) It is exceedingly difficult to see the structures of chloroplasts in Fig. 11 because of the low resolution. Please improve the quality.
(9) L. 121: Please indicate reference(s) for Fv/Fo representing the efficiency of the water-splitting complex.
(10) L. 164: Please indicate “(b)”.
(11) L. 310; “107-116new1, new 2”?
Author Response
The authors report hermetic effects of Cd on metal homeostasis and photosynthesis in clary sage (Salvia sclarea). I have some comments on the manuscript by Adamakis et al as follows:
- One of the purpose of this study is to understand the physiological responses of clary sage, which is tolerant to heavy metals, to a high concentration of Cd, so the title should include the name of plants used in this study, namely, Salvia sclarea (clary sage).
Yes, it should have been mentioned, thus we included the name clary sage in the title.
- Although Fig. 2 and Fig. 3 show the same kind of analysis for several metal accumulation in clary sage plants, why the authors split these data into two figures? I would like to request to combine them into one figure to make the data easy to compare.
We combined Fig. 2 and Fig. 3 (New Fig. 2).
- As for Figs. 4-6 and Figs. 7-9, I suggest the same thing (the data can be combined into one or two figures) as pointed out earlier for Fig. 2 and Fig. 3.
We combined Figs 5 and 6 into one Fig. (New Fig. 4) since in both of them the chlorophyll fluorescence measurements were under the same light intensity (at 220 μmol photons m-2 s–1), but we left out Fig. 4 (New Fig. 3) in which the chlorophyll fluorescence measurements were made in dark-adapted leaves.
We did not combine Figs 7-9 since each of them refers to different results section and by combining them we would create a problem in putting the larger Figure in the appropriate place in the text.
- One of the most important findings of this study should be the hermetic responses of PSII photochemistry to Cd exposure in clary sage as be in the title. However, the negative effects of longer (8-d) Cd exposure on PSII functionality is only shown in the Fig. S2. Moreover, only the pseudocolor images of Fv/Fm and YII are shown without any quantitative data. The authors should show data for 8-d Cd exposure like as the data for 2-d and 5-d Cd exposure.
We have included the supplemental Figure S2 in the main text as Figure 9 with some quantitative data showing the differences after 8 days exposure. Responses of PSII to longer duration exposure with detail quantitative data showing toxicity symptoms have been include in another manuscript that has been submitted for publication and it is cited in the present manuscript (Reference 127).
The results that we wanted to show in this manuscript in detail were those referring to the stimulation of PSII functionality under short duration exposure to a high Cd concentration while the toxicity results of longer duration exposure with the same Cd concentration are presented in detail in another manuscript (Reference 127).
Yet, we included some more new data regarding oxidative stress measurements (New Figures 10 and 11) that were asked by reviewer No 1.
- The authors show that NPQ is increased in short-term Cd treated plants, but no underlying mechanism is investigated and discussed. NPQ can be decomposed mainly into 3 processes, namely qE, qT (or qZ) and qI. Particularly, under stress conditions, distinguishment of energy-dependent qE and photoinhibitory qI is important for understanding which NPQ mechanism is mainly operated under a specific condition. Because the authors claim the importance of NPQ in the Cd-induced hermetic response of PSII photochemistry in clary sage, which actually consists fundamental parts of their study, this kind of data is necessary to understand the underlying mechanisms for the hermetic responses.
Unfortunately, measurements of the 3 NPQ processes, namely qE, qT (or qZ) and qI are not available. In our manuscript we claim that “The concomitant increase in the photoprotective mechanism of non-photochemical quenching of photosynthesis (NPQ) resulted in an important decrease in the dissipated non-regulated energy (ΦNO), modifying the homeostasis of reactive oxygen species (ROS), through a decreased singlet oxygen (1O2) formation”, and that “Thus, when clary sage was exposed to Cd for a short period, tolerance mechanisms were triggered, with PSII photochemistry to be regulated by NPQ in such a way that PSII efficiency to be enhanced.”
- No data for growth data (plant size, fresh weight or dry weight) are shown in the manuscript, but these data are very important to consider a relationship between growth and photosynthetic activity under the stress conditions.
We have included in the manuscript (lines 121-123, and 201-202) data referring to plant growth performance. Cd exposure up to 5 days had not any negative or positive significant influence on plant biomass, but 8 days Cd exposure reduced whole plant biomass by 18% (p<0.05).
(7) Although the authors state that no noticeable disruption was detected in chloroplast ultrastructure, but how was the content of photosynthetic pigments (chlorophyll and carotenoids)? Chlorophyll fluorescence parameters only show the efficiency and do not tell us the total photosynthetic activity. In this context, profiles of pigment content, particularly chlorophyll content because magnesium decreased by the Cd exposure, during the Cd exposure are necessary to estimate functionality of chloroplasts under Cd-exposed stresses.
Such data have been included in our manuscript that has been submitted for publication and it is cited in the present manuscript (Reference 127), thus we cannot present them here.
(8) It is exceedingly difficult to see the structures of chloroplasts in Fig. 11 because of the low resolution. Please improve the quality.
We provided a higher resolution Figure of chloroplast structure (Figure 12 in the new manuscript) as well as a new Figure with disruption of chloroplast structure after 8 days exposure (Supplemental Figure S1).
(9) L. 121: Please indicate reference(s) for Fv/Fo representing the efficiency of the water-splitting complex.
We included one citation for Fv/Fo in line 121 (line 119 in the new manuscript) and following in the text (line 299) two citations for Fv/Fo. This was done in order to avoid an extensive renumbering of references (the first citation was already in the Reference list, but not the second one) that sometimes can result in mistakes.
(10) L. 164: Please indicate “(b)”.
Yes, we indicated it. Thanks for pointing this.
(11) L. 310; “107-116 new1, new 2”?
They were left there by mistake. They indicated two new citations that we wanted to include there and were written as new1 and new 2 in order to remember them in the reference list. Unfortunately, we forget to erase these words later after having included these two citations in the reference list. Thanks for pointing this.
Round 2
Reviewer 1 Report
I found that the revised manuscript has improved substantially with new data of H2O2 and MDA levels, and the authors’ conclusion has become convincing by the revision. I was quite impressed that the authors performed additional experiments during this short period of time.
However, I have to point out a few problems in the revised manuscript as below. I hope the authors will consider modification for them in the next version.
1) I think the newly added sentence in Abstract (L29-32) is misleading and may give wrong impression to readers. The authors mention that the ROS level under 5 days Cd exposure was low, but the data in Figure 10 indicate that the ROS level was increased by 5 days treatment and further increased by 8 days treatment. Since the level was higher than that of control, “a low level of ROS under 5 days Cd exposure” is incorrect. Also the data in Figure 10 do not indicate that the ROS level under 5 days exposure is “beneficial”, which is the authors’ opinion (or discussion) and not experimental result. Therefore, I think it would be better to change these phrases to “ROS level was increased under 5 days Cd exposure, which might be beneficial for triggering defense response.”
2) I’m afraid I cannot understand the latter half of the newly added sentence in L344-346, in which “signaling pathway” is redundant. It seems to me that this sentence mention that "their action in signaling pathway" sometimes antagonize each other. Is it right?
3) “Figure 12” which appear in L357 and L497 should be “Figure 13”. The same goes for “Figure 11” in L493.
Author Response
- I think the newly added sentence in Abstract (L29-32) is misleading and may give wrong impression to readers. The authors mention that the ROS level under 5 days Cd exposure was low, but the data in Figure 10 indicate that the ROS level was increased by 5 days treatment and further increased by 8 days treatment. Since the level was higher than that of control, “a low level of ROS under 5 days Cd exposure” is incorrect. Also the data in Figure 10 do not indicate that the ROS level under 5 days exposure is “beneficial”, which is the authors’ opinion (or discussion) and not experimental result. Therefore, I think it would be better to change these phrases to “ROS level was increased under 5 days Cd exposure, which might be beneficial for triggering defense response.”
Yes, you are right. The sentence you mentioned could give a wrong impression to readers. We modified the respective sentences according to your suggestion (L30-31, L325-326). Thank you for pointing it.
- I’m afraid I cannot understand the latter half of the newly added sentence in L344-346, in which “signaling pathway” is redundant. It seems to me that this sentence mention that "their action in signaling pathway" sometimes antagonize each other. Is it right?
We modified the sentence by deleting the “signaling pathway”(L321, in the new manuscript). Yes, we want to mention that "their action in signaling pathway" sometimes antagonize each other.
- “Figure 12” which appear in L357 and L497 should be “Figure 13”. The same goes for “Figure 11” in L493.
Yes, we corrected these mistakes. However, since a new Figure was included as requested by the other Reviewer the Figure numbers changed. Thank you for your correction.
Reviewer 2 Report
Most of my concerns in the first review have been correctly revised, but still I believe that the chlorophyll content data is necessary for "this" manuscript, because the content of Mg, which is the essential metal component of chlorophyll, is largely decreased by the 5-d Cd treatment. This result makes us assume the possibility that the decreased Mg uptake suppresses chlorophyll biosynthesis or induces chlorophyll degradation to maintain available Mg in cells, which may thereby result in transient surplus of photosynthetic proteins against chlorophylls and improve photosynthetic efficiency temporarily during the short Cd treatment. Actually they have chlorophyll content data, so they should include them in this manuscript.
Author Response
Most of my concerns in the first review have been correctly revised, but still I believe that the chlorophyll content data is necessary for "this" manuscript, because the content of Mg, which is the essential metal component of chlorophyll, is largely decreased by the 5-d Cd treatment. This result makes us assume the possibility that the decreased Mg uptake suppresses chlorophyll biosynthesis or induces chlorophyll degradation to maintain available Mg in cells, which may thereby result in transient surplus of photosynthetic proteins against chlorophylls and improve photosynthetic efficiency temporarily during the short Cd treatment. Actually they have chlorophyll content data, so they should include them in this manuscript.
We included data of chlorophyll a and chlorophyll b content after Cd exposure (new Figure 3), that confirmed your suggestion of a decreased chlorophyll content due to decreased Mg availability (as we mention in L120-121). However, (as we mention in L 352) Mg content in the leaves after 5 days Cd exposure (2778 ± 83 µg g-1) remained higher than the adequate range limit (2000 µg g-1).